# Eosinophilic Esophagitis: Cytokines Expression and Fibrotic Markers in Comparison to Celiac Disease

**DOI:** 10.3390/diagnostics12092092

**Published:** 2022-08-29

**Authors:** Annamaria Pronio, Francesco Covotta, Lucia Pallotta, Rossella Palma, Danilo Badiali, Maria Carlotta Sacchi, Antonietta Lamazza, Carola Severi

**Affiliations:** 1Department of General Surgery and Surgical Specialties ‘Paride Stefanini’, Sapienza University of Rome, 00185 Rome, Italy; 2Department of Translational and Precision Medicine, Sapienza University of Rome, 00185 Rome, Italy; 3Department of Surgery Pietro Valdoni, Sapienza University of Rome, 00185 Rome, Italy

**Keywords:** eosinophilic esophagitis, fibrosis, chronic inflammation, cytokines expression, celiac disease

## Abstract

Introduction: Eosinophilic esophagitis (EoE) is now recognized as the main inflammatory condition that leads to fibrosis, unlike other chronic inflammatory gastrointestinal diseases, such as celiac disease. The aim of our study is to characterize the collagen deposition and cytokine expression involved in the fibrogenic response in patients affected by EoE in comparison to celiac disease. Materials and Methods: Consecutive patients with a clinical suspicion of untreated EoE or active celiac disease were enrolled. In the control group, patients with negative upper endoscopy were included. Total RNA was isolated from biopsy specimens using a commercial kit (SV Total RNA Isolation System, Promega Italia Srl). Quantitative real-time PCR (qRT-PCR) was performed in triplicate using a StepOne™ Real-Time PCR instrument (Thermo Fisher Scientific, Monza, Italy). mRNA encoding for inflammatory molecules: interleukin 4 (IL-4), interleukin 5 (IL-5), interleukin 13 (IL-13), and fibrotic markers: transforming growth factor beta 1 (TGF-β), mitogen-activated protein kinase kinase kinase 7 (MAP3K7), serpin family E member 1 (SERPINE1), were quantified using TaqMan Gene Expression Assays (Applied Biosystems). RESULTS. In EoE, the qPCR analysis showed an increase in all the inflammatory cytokines. Both IL-5 and Il-3 mRNA expression resulted in a statistically significant increase in oesophageal mucosa with respect to the celiac duodenum, while no differences were present in IL-4 expression. TGF-β expression was similar to the controls in the mid esophagus but reduced in the distal EoE esophagus (RQ: 0.46 ± 0.1). MAP3K7 expression was reduced in the mid esophagus (RQ: 0.59 ± 0.3) and increased in the distal esophagus (RQ: 1.75 ± 0.6). In turn, the expression of SERPINE1 was increased in both segments and was higher in the mid than in the distal esophagus (RQ: 5.25 ± 3.9, 1.92 ± 0.9, respectively). Collagen deposition was greater in the distal esophagus compared to the mid esophagus [18.1% ± 8 vs. 1.3% ± 1; *p* = 0.008]. Conclusions: The present study confirms the esophageal fibrotic involution involving the distal esophagus and shows that the inflammatory pathway in EoE is peculiar to this disease and different from other chronic inflammatory gastrointestinal disorders such as celiac disease.

## 1. Introduction

Eosinophilic esophagitis (EoE) has been defined as a chronic, immune/antigen-mediated, esophageal disease characterized histologically by eosinophil-predominant inflammation and clinically by symptoms related to esophageal dysfunction [1]. The clinical manifestations of EoE vary with age. Adults frequently report dysphagia and food impactions, whereas, in younger children, symptoms often include feeding difficulties, gastroesophageal reflux symptoms, and abdominal pain [2]. Dysphagia in solid foods is the most common symptom. Up to 15% of patients undergoing upper endoscopy for dysphagia are found to have eosinophilic esophagitis [3,4,5,6,7].

In the upper gastrointestinal tract, eosinophilic esophagitis is now recognized as the main inflammatory condition that leads to fibrosis, unlike other chronic inflammatory gastrointestinal diseases, such as celiac disease. Dysphagia has been explained as an epiphenomenon of inflammation associated with motor alterations and organ remodeling for collagen deposition.

Adaptive T cell immunity, driven by T helper type 2 (Th2) cells and involving IL-4, IL-5, and IL-13 expression, appears to play a major role in the pathogenesis of EoE and the alteration of esophageal motility [8,9,10,11,12]. IL-4 has several biological roles, including Th2 stimulation, activation, and proliferation (which produce IL-4 and IL-5) [13]. IL-5 is expressed by Th2 cells and is a key mediator in eosinophil activation, eosinophil-induced esophageal remodeling, and collagen deposition [14]. IL-13 promotes the recruitment of eosinophils and the eotaxin-3 secretion [15] and stimulates smooth muscle cell contractility and collagen deposition [16].

Dysregulated innate and adaptive immune responses are major contributors to fibrosis as well as cell-intrinsic modifications in fibroblasts and other structural cells. Switches towards a fibrogenic phenotype (productive) in mesenchymal cells both at the lesion sites (e.g., fibroblasts, smooth muscle cells) and in circulating cells (e.g., circulating fibrocytes and bone marrow progenitors) have been demonstrated [17]. The current opinion is that the activation of tissue-resident fibroblasts is the major source of activated myofibroblasts, although epithelial to mesenchymal transition (EMT), endothelial to mesenchymal transition (EndoMT), or pericyte to myofibroblast transition may play a role in particular circumstances [18]. Myofibroblasts are a population of mesenchymal cells that display a marked increase in the production of fibrillar collagens (types I, III, V, and VI) and other ECM macromolecules coupled with an increase in specific Tissue Inhibitors of Metalloproteinase (TIMPs) production leading to inhibition of ECM-degradative enzymes [19]. The induced differentiation into ECM-producing active fibroblasts, together with a relative reduction of the degradation of the extracellular matrix components (ECM; Collagen, fibronectin), ends in the exaggerated and uncontrolled accumulation of extracellular matrix (ECM) macromolecules in the affected tissues and the replacement of normal tissues with fibrotic tissue [20,21,22,23]. 

TGF-β is a potent inducer of myofibroblasts either through the activation of quiescent fibroblasts or through the phenotypic conversion of various cell types into activated myofibroblasts [24,25,26]. A number of TGF-β signaling pathways have been identified, and TGF-β-activated kinase 1 (TAK1 or MAP3K7) is a major upstream signaling molecule in TGF-β1-induced type I collagen and fibronectin expression through activation of the MAPK kinase (MKK)3-p38 and MKK4-JNK signaling cascades, respectively [27]. Further, a key role in contributing to fibrosis is played by an excessive Serpine 1 production that contributes to the excessive accumulation of collagen and other ECM proteins [28].

Even though cell-intrinsic changes in important structural cells that can perpetuate the fibrotic response by regulating the differentiation, recruitment, proliferation, and activation of myofibroblasts have been deeply described in several diseases, the interplay between cytokines and EoE fibrosis has not completely been clarified. The aim of our study is to characterize the collagen deposition and cytokine expression involved in the fibrogenic response in patients affected by eosinophilic esophagitis in comparison to celiac disease, a non-fibrosis-associated chronic inflammatory gastrointestinal disease. The secondary outcome is to define whether the differential expression of pro-inflammatory cytokines corresponds to different fibrotic progressions of the disease.

## 2. Materials and Methods

Consecutive patients who attended the Endoscopy Unit of the Academic Hospital Umberto I of Rome with a clinical suspicion of untreated EoE or active celiac disease were enrolled. The diagnostic criteria of EoE were: 1. symptoms related to esophageal dysfunction (dysphagia, retrosternal heartburn, regurgitation, bolus impact) and 2. eosinophil-predominant inflammation on esophageal biopsy, characteristically consisting of a peak value of ≥15 eosinophils per high power field (HPF). For each patient, clinical data were recorded, and a pharmacological history, in particular regarding the intake of steroids, was obtained. Ethical approval for this study was obtained from AOU Policlinico Umberto 1—Sapienza Università di Roma (approval number: 6244).

The diagnosis of celiac disease was established according to the last European Society for the Study of Coeliac Disease (ESsCD) guidelines [29]. In the control group, patients with negative upper endoscopy results were included. 

The exclusion criteria were: age < 18 years, presence of neoplastic lesions, previous radiation treatments, and previous upper GI surgery.

All enrolled patients underwent esophagogastroduodenoscopy with multiple biopsies. All the procedures were performed under conscious sedation (Ipnovel + Xylocaina spray). During endoscopy, biopsies were obtained from the distal esophagus, 3 cm above the squamocolumnar junction, as well as the mid esophagus. In patients with suspected active celiac disease, two biopsies of the bulb and two biopsies of the second duodenal portion were performed. The accuracy of these biopsies was improved with staining techniques by using specific Endokits consisting of cellulose nitrate filter membranes to obtain the correct orientation of the sampling. In order to perform both histological examination and genetic analysis, biopsies of each portion were preserved respectively in 4% formalin and RNA samples later at −80 °C.

Biopsy specimens placed in 4% formalin were included in paraffin and subsequently cut by microtome and stained with Hematoxylin-Eosin (simple histology) and Picrosirius red staining for the collagen stain (Figure 1). To quantify the deposition of collagen, a histomorphometric method was used in the gastrointestinal mucosa [30]. The images were acquired by a Leitz DyRB (Leica) optical microscope at (×2.5) magnification and analyzed by IAS 2000 software, Delta Systems, by two expert pathologists. Collagen deposition extent was expressed as the percentage of tissue staining (positive tissue area (mm^2^) × 100/total section area (mm^2^)). For each biopsy sample, only the sections including the lamina propria were considered.

## 3. Total RNA Extraction

Total RNA was isolated from biopsy specimens using a commercial kit (SV Total RNA Isolation System, Promega Italia Srl), and the integrity of the RNA was confirmed by electrophoresis on 1% agarose gel and ethidium bromide staining (0.1 mg/mL). Optical density at 260 nm was used to evaluate total RNA concentration. RNA samples were stored at −80 °C.

## 4. Quantitative Real-Time PCR

500 ng of RNA was reverse-transcribed in a 20 μL reaction tube using a high-capacity cDNA reverse transcription kit (Applied Biosystems, Monza, Italy). Quantitative real-time PCR (qRT-PCR) was performed in triplicate using a StepOne™ Real-Time PCR instrument (Thermo Fisher Scientific, Monza, Italy) following the standard protocol. mRNA encoding for inflammatory molecules: interleukin 4 (IL-4), interleukin 5 (IL-5), interleukin 13 (IL-13), and fibrotic markers: transforming growth factor beta 1 (TGF-β), mitogen-activated protein kinase kinase kinase 7 (MAP3K7), serpin family E member 1 (SERPINE1), were quantified using TaqMan Gene Expression Assays (Applied Biosystems), with specific human primers selected from the Applied Biosystem database (IL-4: Hs00174122_m1; IL-5: Hs99999031_m1; IL-13: Hs00174379_m1; TGF-β: Hs00998133_m1; MAP3K7: Hs00177373_m1; SERPINE1: Hs00167155_m1; ACTB: Hs01060665_g1). The amount of the target gene was normalized to β-actin (ACTB). The 2^−ΔΔCt^ method was applied to determine the relative changes in gene expression levels (RQ). The results have been analyzed with SDS 2.1 Applied Biosystems software.

## 5. Statistical Analysis

The descriptive statistic was expressed as the mean ± SE. GraphPad Prism 8 software has been used for statistical analysis (IBM SPSS, INC., Chicago, IL, USA). The statistical significance has been calculated by Mann–Whitney’s nonparametric independent sample test. The nonparametric correlations were calculated using the Spearman correlations. A *p*-value lower than 0.05 was considered significant.

## 6. Results

Seventeen patients were enrolled. Of these, five had a diagnosis of EoE, five had a diagnosis of celiac disease, and seven were control cases: four for esophageal and three for duodenal biopsies. Demographic characteristics are reported in Table 1. 

## 7. mRNA Expression Analysis

(a)
**Inflammatory cytokines**


In EoE mucosal samples obtained from the middle and distal esophagus, the qPCR analysis showed an increase in all cytokines, both in the mid and distal esophagus, compared with their respective controls. In particular, in the mid-esophagus, a significant increase was observed both in IL-5 (RQ: 327.22 ± 250.7) and IL-13 (RQ: 764.66 ± 461.4). In this region, a 10-fold increase was also observed in IL-4, even if it was not significant. In the distal esophagus, a significant increase was observed in IL-4 (RQ: 5.85 ± 2.4) and IL-5 (RQ: 276.31 ± 231.4), with a lower pronounced expression of IL-13 than in the mid esophagus (RQ: 50.9 ± 32.2), but still greater than control. The differences in the mucosal gene expression of inflammatory cytokines between EoE and celiac disease were then compared using samples from celiac patients, regardless of the region of the biopsy sample, as a reference to the analyzed gene expression of all samples of EoE, without distinction between the mid and distal regions. Both IL-5 and Il-3 mRNA expression resulted in a statistical increase in oesophageal mucosa with respect to the celiac duodenum, while no differences were present in IL-4 expression. (Figure 2).

No increase in the three cytokines was observed in the cytokine expression of celiac tissue both in the DII and Bulb regions of the duodenum. In the bulb, a light increase, not significantly different from the control, was observed in the expression of the three cytokines (RQ IL4: 1.68 ± 0.5; IL-5: 1.2 ± 0.4; IL-13: 1.13 ± 0.5) that in turn were slightly under-expressed in DII (RQ IL4: 1.03 ± 0.8; IL-5: 0.55 ± 0.3; IL-13: 0.46 ± 0.4; Figure 3).

(b)
**Pro-fibrogenic mediators**


Regarding the expression of fibrotic mediator genes in EoE, some regional differences were observed, mainly concerning TGF-β and MAP3K7 expression. TGF-β expression was similar to the control in the mid-esophagus but reduced the distal EoE esophagus (RQ: 0.46 ± 0.1). MAP3K7 expression was reduced in the mid-esophagus (RQ: 0.59 ± 0.3) but increased in the distal esophagus (RQ: 1.75 ± 0.6). In turn, the expression of SERPINE1 was increased in both segments and was higher in the mid than the distal esophagus (RQ: 5.25 ± 3.9, 1.92 ± 0.9, respectively). When the expression of fibrotic markers was compared between EoE and celiac disease, using the latter as a reference, their expression was higher in EoE (Figure 4).

In celiac disease, no differences in markers expressions from the control were observed in the duodenal bulb, while the down-expression of all fibrotic markers was observed in the DII mucosal region (RQ: TGF-β 0.36 ± 0.1, MAP3K7 0.22 ± 0.04, SERPINE1 0.2 ± 0.05; Figure 5).

(c)
**Collagen deposition quantification**


Collagen deposition evaluation required the identification of the lamina propria and was evaluable only in 50% of the bioptic specimens. Twenty biopsy specimens from patients with EoE and celiac disease (10 from the distal esophagus and 10 from the bulb and II duodenal portions) were collected. Only in 10/20 (50%) specimens, the lamina propria was identified. Specifically, in patients with EoE, the lamina propria was detectable in 3/5 samples of the distal esophagus and 2/5 samples of the mid-esophagus (n = 5); in patients with celiac disease, the lamina propria was identifiable in 3/5 bulb samples and 2/5 samples of the II duodenal portion. The collagen deposition was greater in the distal esophagus compared to the mid-esophagus (18.1 ± 8% vs. 1.3 ± 1%; *p* = 0.008). No significant correlations were found between the amount of fibrosis and the inflammatory mediator expression.

In patients with celiac disease, the presence of duodenal collagen deposition was seen in 7.0 ± 4%. In the distal esophagus of patients with EoE, the presence of collagen was significantly higher than that of celiac patients (18.1 ± 8% vs. 7.0 ± 4; *p* = 0.046). No significant differences were identified regarding the presence of collagen deposition between the mid esophagus of patients with EoE and celiac disease patients (1.3 ± 1% vs. 7 ± 4%; *p* = NS). 

## 8. Discussion

EoE is a relatively rare pathology, with an estimated prevalence of one to five patients per 10,000 habitants, and the incidence of EoE appears to be increasing [31,32,33,34,35] due to the greater recognition of the disorder. The pathogenesis of EoE is not completely understood but involves genetic, environmental, and host immune system factors. The present study demonstrated that the inflammatory pathways involved in EoE are peculiar to the disease and are different from those involved in other GI inflammatory diseases, such as celiac disease. Specifically, we reported the overexpression of specific cytokine mediators, IL-13, IL-4, IL-5, and the pro-fibrotic mediator SERPINE 1 in mucosal samples of EoE. Both inflammatory and pro-fibrogenic markers were overexpressed in comparison to celiac disease.

Our results confirm the previous finding of the increased secretion of IL-5 and IL-13 in EoE compared with control samples [8]. However, it is of great interest to identify the different cytokine pathways in two different diseases with chronic inflammation, such as celiac disease and eosinophilic esophagitis. Among inflammatory proteins, the more notable increased expression was observed for IL-5 and IL-13. IL-5 plays a key role in the activation, maturation, and migration of eosinophils [36] and is also involved in the mechanism of eosinophilic-mediated esophageal remodeling and collagen deposition [37]. It has been previously shown that IL-5 could induce eosinophilic esophagitis in animal models, and the administration of anti-IL-5 antibodies seems to block this process [38]. Some randomized clinical trials have also evaluated the efficacy of IL-5 inhibitors in patients with EoE [39,40,41], and it was shown that, despite a significant reduction in eosinophil infiltration, its effectiveness on symptoms was limited.

IL-13 is known to be involved in atopic diseases stimulating eosinophil chemotaxis, eotaxin-3 secretion, collagen deposition, and smooth muscle cell contractility [15,16]. This cytokine has already been reported to be over-expressed in mucosal samples of patients affected by EoE with respect to healthy controls [9]. In murine models, the intratracheal administration of IL-13 induced eosinophilic esophagitis with a dose-dependent trend [42]. This process was arrested by the administration of anti-IL-13 monoclonal antibodies. Recently, it has been shown [43] that the administration of anti-IL-13 monoclonal antibodies resulted in an improvement in both endoscopic and histological findings in 99 patients with EoE. It was not associated, however, with significant symptom improvement. 

IL-4 is an important regulator of adaptive and humoral immunity, and it seems to be involved in the isotypic switch in IgE production [44]. It is also known to induce Th2 stimulation [13]. In the present study, this cytokine was not particular to the inflammatory pathway of EoE but was likely related to Th2-driven inflammation since no differences in its expression were found between EoE and celiac disease. 

Rieder et al. [8] previously showed the dual direct and indirect involvement of eosinophil-derived mediators in the fibrosis and dysmotility that characterize EoE. They demonstrated that these mediators derive not only from the epithelial layer but also from the submucosa and muscles, confirming the transmural nature of the disease.

We also found that the expression of fibrotic markers, in particular SERPINE1, was increased in both the mid and distal esophagus. This finding emphasizes the interaction of eosinophils with fibroblasts and muscle cells in which TGFβ1 appears to be a dominant driver, as has been previously shown [8]. In our study, we found that TGF-β expression was similar to controls in the mid esophagus, which was reduced in the distal EoE esophagus. Despite this, when the expression of fibrotic markers was compared between EoE and celiac disease, using the latter as a reference, their expression was higher in EoE.

In the hypothesis that persistent and/or recurrent inflammatory processes are associated with the switch of mesenchymal cells (e.g., fibroblasts, smooth muscle cells) to a fibrogenic (productive) phenotype, the identification of different cytokines pathways in two different diseases such as celiac disease and eosinophilic esophagitis should be very useful. The cellular differentiation into active ECM-producing myofibroblasts is an essential event in the fibrogenic process [45]. Therefore, it is possible that the over-expression of these mediators may contribute to eosinophil increases and esophageal remodeling, resulting in reduced esophageal distensibility that could explain the onset of dysphagia. In patients with eosinophilic esophagitis, the origin of the dysphagia, except for patients with esophageal stenosis, is not completely clarified. 

In the present study, it was confirmed that eosinophilic esophagitis is characterized by marked fibrotic tissue involution. The fibrotic process appears to have an irregular distribution and involves the distal esophagus more than the mid, confirming previous data from Wang et al. [46] that observed more fibrosis in the distal third rather than in the proximal esophagus. This should be explained by the role of gastric acid reflux or by a hypothetically greater stagnation of the bolus at the level of the lower esophageal sphincter (LES) and consequently increased tissue inflammation. Although in this study, no clear correlation between a specific cytokine pathway and the presence of fibrosis was found, based on gene expression analysis, it is likely that the different histological aspects are to be referred to as different inflammatory pathways.

The lack of correlation might be related to the reduced number of specimens available for the study of fibrosis. Indeed, the presence of the lamina propria, where fibrosis could be evaluated, was identified in 50% of the biopsy samples. This result is comparable to the study by Wang et al. [46], where the lamina propria was evaluable in 42% of biopsy specimens. Therefore, it would be recommended to perform more biopsies in future studies to be sure to identify the lamina propria and perform more accurate histological examinations.

Pro-fibrogenic markers were highly expressed in the EoE mucosa with respect to the duodenal mucosa of celiac patients. Among the pro-fibrogenic markers analyzed, SERPINE 1 appeared to be homogeneously increased both in the mid and distal esophagus. This marker has already been demonstrated to contribute to the pro-fibrotic trend in EoE [47]. The increase in SERPINE 1 in both the EoE mid and distal esophagus compared with their specific controls and celiac samples supports the presence of fibrosis only in EoE. However, TGF-β expression, known to be an important tissue fibrosis mediator and to induce myofibroblasts to synthesize extracellular matrix [48], has not been found to be increased in EoE with respect to controls, even if highly expressed in comparison to celiac disease. TGF-β plays a central role in the fibrogenesis process. Nevertheless, TGF-β is an important immunoregulatory cytokine involved in the activation of regulatory T lymphocytes [49], and cytokine expression inhibition could be correlated with an increased inflammatory response [45,50].

A number of TGF-β signaling pathways have been identified, and TGF-β-activated kinase 1 (TAK1 or MAP3K7) is a major upstream signaling molecule in TGF-β1-induced type I collagen and fibronectin expression through the activation of the MAPK kinase (MKK)3-p38 and MKK4-JNK signaling cascades, respectively [27,51,52]. After all, MAP3K7, which is increased in the distal esophagus, can also be activated by various stimuli, including pro-inflammatory cytokines [53].

In conclusion, the present study confirms esophageal fibrotic involution involving the distal esophagus and shows that the inflammatory pathway in EoE is different from other chronic inflammatory gastrointestinal disorders such as celiac disease. Based on our data, specific pro-inflammatory cytokines, in particular, IL-5 and IL-13, seem to play key roles. Further, the over-expression of pro-fibrogenic markers in this chronic inflammatory condition could explain the fibrotic progression and esophageal remodeling, resulting in reduced organ compliance and esophageal motility alterations. The main limitation of this study is the small sample size due to the low prevalence of this disease. However, further studies, by using recently validated specific histological scores for the evaluation of fibrosis, aiming to identify a wider cytokine pathway associated with fibrosis development in eosinophilic esophagitis are necessary in order to recognize specific mediators that could be targeted as potential therapies.

## Figures and Tables

**Figure 1 diagnostics-12-02092-f001:**
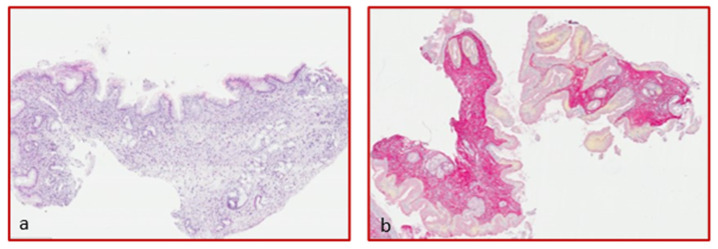
Hematoxylin-eosin staining of an esophageal specimen (**a**) and sirius red (**b**) for collagen quantification (magnification ×4).

**Figure 2 diagnostics-12-02092-f002:**
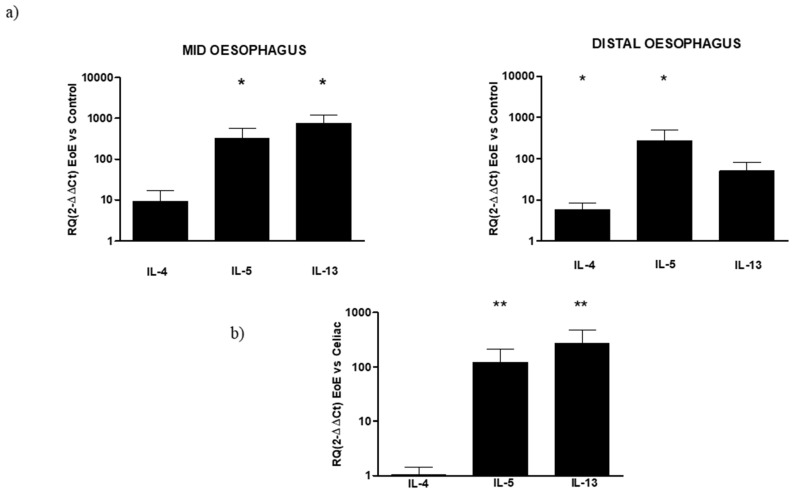
(**a**) Relative expression of mRNA encoding for IL-4, IL-5, and IL-13 in mucosal tissue from the mid and distal esophagus. Data are determined by qPCR (2^−ΔΔCT^ vs. control). Data are expressed as the mean ± SE of four to five experiments, * *p* ≤ 0.05 in EoE mid or distal vs. respective control. (**b**) Relative expression of mRNA encoding for IL-4, IL-5, and IL-13 in mucosal tissue from the EoE esophagus and celiac duodenum. Data are determined by qPCR (2^−ΔΔCT^ EoE vs. celiac). Data are expressed as mean ± SE of eight to 10 experiments, ** *p* ≤ 0.01 in EoE vs. celiac.

**Figure 3 diagnostics-12-02092-f003:**
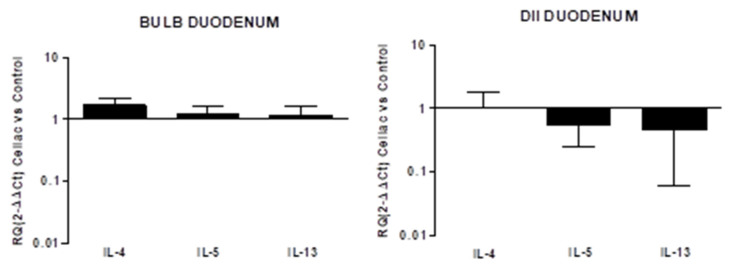
Relative expression of mRNA encoding for IL-4, IL-5 and IL-13 in mucosal tissue from DII and bulb duodenum. Data are determined by qPCR (log 10 2^−ΔΔCT^ vs control). Data are expressed as mean ± SE of 4 experiments.

**Figure 4 diagnostics-12-02092-f004:**
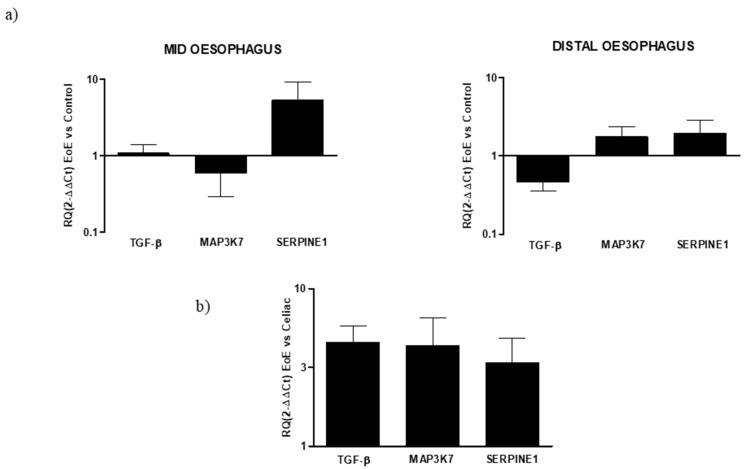
(**a**) Relative expression of mRNA encoding for TGF-β, MAP3K7, and SERPINE1 in mucosal tissue from mid and distal esophagus. Data were determined by qPCR (log 10 × 2^−ΔΔCT^ vs. control). Data are expressed as the mean ± SE of four to five experiments. (**b**) Relative expression of mRNA encoding for TGF-β, MAP3K7, and SERPINE1 in mucosal tissue from the esophagus and duodenum. Data are determined by qPCR (2^−ΔΔCT^ EoE vs. celiac). Data are expressed as the mean ± SE of eight to 10 experiments.

**Figure 5 diagnostics-12-02092-f005:**
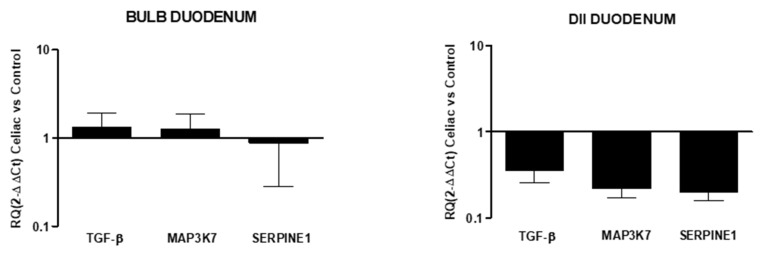
Relative expression of mRNA encoding for TGF-β, MAP3K7, and SERPINE1 in mucosal tissue from the DII and bulb duodenum. Data are determined by qPCR (log 10 2^−ΔΔCT^ vs. control). Data are expressed as the mean ± SE of four experiments.

**Table 1 diagnostics-12-02092-t001:** Demographic characteristics and clinical symptoms.

	EoE	Celiac Disease	Control (Oesophagus)	Control (Duodenum)	*p*-Value
Patients(N)	5	5	4	3	
Biopsy specimens (N)	20	20	16	12	
Sex(M/F)	4/1	0/5	1/3	1/2	0.01
Age mean ± SE	36.4 ± 6	42.2 ± 8	29.7 ± 2	37.6 ± 4	0.44
Caucasian	5 (100%)	5 (100%)	4 (100%)	4 (100%)	1
Clinical symptoms					
Dysphagia	5 (100%)	0	0	0	0
Bolus impact	3 (60%)	0	0	0	0.04
Steroid in the last 4 weeks	0	0	0	0	1

## Data Availability

Not applicable.

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
