# Peer review of "Eosinophilic Esophagitis: Cytokines Expression and Fibrotic Markers in Comparison to Celiac Disease"

_diagnostics, 2022, doi:10.3390/diagnostics12092092_

Round 1
Reviewer 1 Report
General comments:
This manuscript reports on PCR data on type 2 immunity cytokines and some so-called fibrotic markers on samples from patients with eosinophilic esophagitis (EoE) and compared to celiac disease. The number of subjects in each group very low and the data are relatively limited, and the text including the conclusions (and also the title) appear somewhat exaggerated.
Major comments:
- The title claims correlation between cytokine expression and esophageal fibrosis. However, there are no fibrosis data in the manuscript. In order to claim such a correlation, the authors should use and report on the EoEHSS (EoE Histological Scoring System) on their samples and its subscores, including the lamina propria fibrosis subscore, or any similar system.
- Related to point No. 1: Have the authors scored their patients and samples according to the established system, including the EoEHSS, EEsAI (EoE Activity Index) and EREFS (Endoscopic Reference Score), or any other similar systems?
- Table 2: Please provide p values for these data. In the primary statistical analysis of these data, the n should be the number of subjects, not the number of biopsies. A secondary question is what the variation among the specimens from one subject was? The table should also be expanded and ideally include data on age of onset or time since symptom onset, comorbidities, allergic and food sensitivities, severity (e.g., dilations and emergency department visits).
- Please explain more if and how these data are novel compared to the literature, particularly in comparison with reference No. 8 (as mentioned, e.g., on line 259).
- What proteins may be expressed at a higher level in celiac disease than in EoE, may it be factors related to autoimmunity?
- The EoE patients are called “suspect of untreated”. Had they really not received any treatment, including omeprazole?
- Table 1 info could be mentioned in the text. Figures 2-4 could be combined to one panel with parts A, B, etc. Same for figs. 5-7.
- Overall, the text can be shortened, with the possible exception of expanding the part of the Introduction that deals with background on celiac disease.
Minor comment:
- Line 48: EoE is called a “model” here, but it is a real disease. A model would be for instance, an EoE model in mice. Please rewrite.
Reviewer 2 Report
The study explores the cytokines and collagen in esophageal biopsies in individuals with eosinophilic esophagitis (EoE) and compares them to those in the bulb and second duodenal in patients with celiac disease. The sample and methods are well described. It is not clear why celiac disease was chosen as a comparison as a inflammatory bowel disease. While this is an interesting, small study of the inflammatory pathways in EoE showing a difference between celiac disease, it cannot be said based on this study that these markers are unique to EoE.
The title is misleading as there are not correlations and the comparison to celiac disease is not noted in the title.
The abstract and conclusion note that the inflammatory pathway is peculiar to EoE but based on the study one can only say that it is not the same as celiac disease.
The discussion emphasizes the inflammatory pathway to fibrosis in EoE which seems to be the focus of the paper. There is little discussion of the comparison to celiac disease.
The study is small and noted as a limitation by the authors.
Round 2
Reviewer 1 Report
General comments:
This revised manuscript appears to basically have addressed previous comments. However, the pdf file of the revised manuscript one can now download has some layout issues. It’s very difficult to see the revised Table 1 (one sees the previous version of the table more clearly). For instance, one cannot really read the added p values in the table. Also, it is difficult, and possibly more difficult than before, to see the text on the y axes in Figures 2-7. Please provide a new pdf version with clearly readable Table 1 and clearly readable text in all graphs.
Author Response
Thank you very much for your comments and thanks for noticing this problem in the layout. We tried to save the file in a pdf format again. However, the PDF format was optional. You can find the figures uploaded in Tif format according to the journal rules.